# Identification of *Aethina tumida* Kir Channels as Putative Targets of the Bee Venom Peptide Tertiapin Using Structure-Based Virtual Screening Methods

**DOI:** 10.3390/toxins11090546

**Published:** 2019-09-19

**Authors:** Craig A. Doupnik

**Affiliations:** Department of Molecular Pharmacology & Physiology, University of South Florida College of Medicine, Tampa, FL 33612, USA; cdoupnik@health.usf.edu

**Keywords:** bee venom, bioinformatics, computational docking, homology modelling, ion channel structure, protein–peptide interactions, tertiapin, venom peptides, virtual screening, small hive beetle

## Abstract

Venoms are comprised of diverse mixtures of proteins, peptides, and small molecules. Identifying individual venom components and their target(s) with mechanism of action is now attainable to understand comprehensively the effectiveness of venom cocktails and how they collectively function in the defense and predation of an organism. Here, structure-based computational methods were used with bioinformatics tools to screen and identify potential biological targets of tertiapin (TPN), a venom peptide from *Apis mellifera* (European honey bee). The small hive beetle (*Aethina tumida* (*A. tumida*)) is a natural predator of the honey bee colony and was found to possess multiple inwardly rectifying K^+^ (Kir) channel subunit genes from a genomic BLAST search analysis. Structure-based virtual screening of homology modelled *A. tumida* Kir (*at*Kir) channels found TPN to interact with a docking profile and interface “footprint” equivalent to known TPN-sensitive mammalian Kir channels. The results support the hypothesis that *at*Kir channels, and perhaps other insect Kir channels, are natural biological targets of TPN that help defend the bee colony from infestations by blocking K^+^ transport via *at*Kir channels. From these in silico findings, this hypothesis can now be subsequently tested in vitro by validating *at*Kir channel block as well as in vivo TPN toxicity towards *A. tumida*. This study highlights the utility and potential benefits of screening in virtual space for venom peptide interactions and their biological targets, which otherwise would not be feasible.

## 1. Introduction

Venoms from *Hymenoptera* and other venomous species are comprised of diverse mixtures of proteins, peptides, and small molecules [1,2]. The composition of these “venom cocktails” are species-dependent, where the concentration of individual toxin components can vary by age, sex, diet, and different environmental conditions [3]. Envenomation efficacy is largely driven by the adaptive evolutionary pressures that select for certain venom genes based on their ability to either: (1) aid in predation, (2) aid in defense from predation, and/or (3) aid in reproductive health and survival (e.g., anti-microbial and anti-parasitic activity) [4].

One of the well-established molecular targets of venom-derived toxins are the ion channels expressed in natural prey and predators. As potent modulators of ion channel function during envenomation, venom-derived peptides contribute to the overall venom response by causing a variety of physiologic responses including paralysis for host capture and pain for defense from predators [5]. The atomic-level structural details that have emerged for both venom-derived peptides and several ion channels have enabled structure-based computational screening techniques to be deployed for identifying potential target effectors for venom components in silico [6]. These methods are also helping to guide rational peptide design efforts to re-engineer venom peptides for potential drug development purposes [7,8,9].

While most efforts have focused on molecular targets of venom components in laboratory mammals and human tissues, understanding the biological targets of venom-derived peptides encountered in nature is essential for providing a comprehensive understanding of the evolutionary adaptations that drive target selectivity and affinity, and thereby contribute to overall venom efficacy. Such an insight can also be valuable in identifying potential off-target effects in humans where venom-derived peptides are being developed for both therapeutic and non-therapeutic basic research applications [10].

Here, the use of structure-based virtual screening techniques was deployed to search and identify putative ion channel targets using a small venom peptide from *Apis mellifera* (*A. melliferra*) venom called tertiapin (TPN). Venom produced by female worker bees is used primarily for individual defensive purposes, and to protect the colony from various invertebrates as well as vertebrate predators and/or pests. The updated genome assembly of *A. melliferra* (Amel_4.5) with venom transcriptomic analysis indicates over 100 proteins and peptides are present in honey bee venom to collectively mediate these responses [11]. TPN is a 21-amino acid peptide that represents a relatively small fractional component (0.1%) of the total protein content of bee venom [12], and was serendipitously discovered in 1998 to bind and block a subset of mammalian Kir channels [13,14]. The evolution-driven biological target(s) of TPN encountered in nature, however, is currently not established. Here, a hypothesis-driven in silico screening strategy is presented that supports insect Kir channels as potential natural targets for TPN-mediated channel block for bee colony defense from the small hive beetle. This novel finding highlights how the application of structure-based computational tools such as molecular docking can create new research directions by demonstrating mechanistically feasible and testable hypotheses on the adaptive pressures that drive interaction of venom peptides with their natural biological targets.

## 2. Results

### 2.1. Reliability of Molecular Docking of TPN to Kir Channel-Modelled Structures 

The rat Kir1.1 channel is the prototypical TPN-sensitive Kir channel, having an IC_50_ of ~2 nM for functional channel block [14]. Interestingly, however, the human Kir1.1 channel isoform is insensitive to TPN block due primarily to two key amino acid differences in the outer vestibule of the Kir1.1 channel where TPN is known to bind [15,16]. Previous computational docking studies of NMR-derived TPN ensemble conformers (PDB ID 1TER.pdb [17]) to a homology-modelled rat Kir1.1 channel identified a favored TPN conformer having a pore-blocking binding pose where the positively charged C-terminal lysine side chain of TPN inserts and occupies the S1 K^+^ binding site in the channel selectivity filter [7]. Moreover, the computational docking scores reliably differentiate TPN interactions among homology-modelled Kir1.1 isoforms known to be sensitive (rat) and insensitive (human and zebrafish) to TPN_Q_ channel block [7]. Thus, structure-based interactions between TPN and rat Kir1.1, derived from computational docking methods and validated in vitro, are thought to represent the high-affinity bound and blocked state of the channel [13] (but also see Reference [18]).

To further test the reliability of this approach, the same molecular docking protocol used for the homology-modelled rat Kir1.1 channel was applied to crystal structures of the TPN-sensitive murine Kir3.2 channel homo-tetramer [19,20]. Mammalian Kir3.2 channel subunits are expressed in neuronal and endocrine cells and assemble primarily as hetero-tetrameric channels with different Kir3.x subunits, and function to inhibit membrane excitability [21]. When heterologously expressed in *Xenopus* oocytes, the homo-tetrameric Kir3.2 construct used for X-ray crystallography yields channels functionally blocked by the oxidation-resistant TPN_Q_ variant peptide in a manner similar to hetero-tetrameric Kir3.1/3.2 channels [20,22,23,24]. Thus, docking TPN to the Kir3.2 crystal structure precludes homology modelling of the Kir channel. 

Shown in Figure 1, rigid-body docking of TPN conformers to the Kir3.2 channel crystal structures was dependent on the NMR-derived TPN conformer used for docking, similar to that observed with TPN docking to the homology-modelled rKir1.1 channel [7]. For the Kir3.2 channel, the TPN conformer that yielded the highest docking score for both the closed-state and pre-opened-state conformations was TPN-13.

To further characterize the structural similarity among the 21 TPN conformers, hierarchical clustering analysis of the calculated pairwise alpha-carbon amino acid backbone RMSD values for the NMR ensemble structures was performed (Figure 2). The RMSD cluster analysis identified 4 distinct sub-conformations for TPN, where conformers TPN-12 and TPN-13 belonged to the same sub-conformation cluster group and were closest in structural similarity among the 21 conformers (Figure 2A). As previously reported, the TPN-12 conformer yielded the top-ranked docking score for the homology-modelled rat Kir1.1 channel [7] and represented the second best docking score for Kir3.2 (see Figure 1). Thus, the TPN docking results to Kir3.2 channels in both closed and pre-opened conformations are in good agreement with the previously characterized rat Kir1.1 docking study and support a favored role for the TPN sub-conformation state represented by the cluster group that includes TPN-12 and TPN-13 conformers. One of the major structural contributors mediating the different TPN sub-conformations was the orientation of the C3–C14 disulfide bond that contributes to the overall tertiary peptide structure and orientation of the surface exposed side chains (Figure 2B). 

Surface renderings of TPN-13 docked to the Kir3.2 channel are shown in Figure 3 where the entire TPN peptide structure is shown to be bound deep within the channel outer vestibule and plugs the channel like a “cork in a bottle”. The TPN peptide interacts in an asymmetric manner making different contacts with the four identical Kir3.2 subunits. However, similar to the modelled Kir1.1 channel, the C-terminal TPN K21 side chain occupies the central channel pore with other TPN residues making contact with Kir3.2 vestibule residues including the Kir3.2 turret structures.

### 2.2. Virtual Screening for TPN-Interacting Kir Channels 

Given the reproducible outcomes for TPN docking to homology-modelled rKir1.1 channel and the two mKir3.2 channel crystal structures, homology models for 14 different mouse Kir channel isoforms (Figure 4A) were constructed analogous to rKir1.1 and then screened in silico for docking interactions with the TPN-12 conformer. The homology-modelled Kir channels were all homo-tetramers, and the only Kir channel not examined was Kir2.3 which contained a significantly larger extracellular turret loop region that precluded reliable modelling and assembly of the Kir2.3 homo-tetramer using the Swiss-Model homology-modelling program (see Materials and Methods section).

The ranked TPN-12 docking score profile obtained from the virtual screen of mouse Kir channel structures is shown in Figure 4B, where the scoring algorithm is weighted for receptor–ligand shape complementarity, electrostatic contacts, and van der Waal forces [25]. As reported previously, the top-ranked docking scores begin to rapidly diminish within the top ten scored complexes. The average score for the top 5 complexes for each Kir channel is shown in Figure 4C. In agreement with functional studies, three mouse Kir channels with the highest docking scores were channels known to be sensitive to functional TPN_Q_ block, namely Kir1.1, Kir3.2, and Kir3.4. Unexpectedly, however, the Kir channel having the second highest docking score was Kir4.1. When examining the top-ranked docking scores, the order for energetically favored interactions with TPN-12 was Kir1.1 > Kir4.1 > Kir3.4 > Kir3.2 > Kir3.1. The Kir3.1 docking scores were comparable to the TPN-insensitive Kir2.1 channel, which is in agreement with the insensitivity of homomeric mutant Kir3.1 channels to TPN_Q_ block [26]. Moreover, the other seven Kir channels with lowest TPN docking scores are also known to be insensitive to TPN_Q_ block (i.e., Kir2.x, Kir6.2, and Kir7.1) and therefore establish a baseline profile for Kir channels insensitive to TPN [27]. These initial findings from the virtual screen across mouse Kir channels largely agree with known Kir channel TPN sensitivities, but suggest the modelled Kir4.1 channel outer vestibule structure presents a viable receptor target for TPN interactions comparable to known TPN-sensitive Kir channels.

### 2.3. Interface Analysis of TPN-Docked Kir Channels

Given the unexpected high TPN docking score to the homology-modelled Kir4.1 channel, refined docking of TPN to Kir4.1 was performed next to compare the predicted Kir channel–TPN interface contacts using PISA [28]. Shown in Figure 5, the TPN interface “footprint” on the Kir4.1 channel vestibule was similar to those on both the TPN-sensitive rat Kir1.1 and mouse Kir3.2 channels, where three major subunit contact sites within the Kir channel vestibule were observed. These interface “hot spots” corresponded to: (1) the turret region, (2) a “ring” region located along the wall of the channel vestibule, and (3) the pore entry region. Most of the exposed residue side chains of TPN participate in the predicted asymmetric binding interface in a Kir subunit-dependent manner. For mKir4.1, the predicted formations of hydrogen bonds were fewer, sharing some conserved sites in the channel turret and pore regions but absent in the mid-level ring region where mKir4.1 lacks an equivalent acidic reside present in rat Kir1.1 and mouse Kir3.2. The docking pose for TPN-12 within the Kir4.1 vestibule is shown in Figure 6, indicating a similar general orientation where the TPN C-terminal K21 side chain was positioned and inserts into the channel pore.

### 2.4. Testing TPN_Q_ Block of Kir4.1 Channels Expressed in Xenopus Oocytes

To test whether TPN functionally blocks mKir4.1 channels, the mouse Kir4.1 isoform was expressed in *Xenopus* oocytes and K^+^ currents recorded before and during 100 nM TPN_Q_ application. Shown in Figure 7, K^+^ currents produced by expressed Kir4.1 channels were insensitive to 100 nM TPN_Q_, a concentration that blocks nearly 100% of the rKir1.1 channel current [7,14]. The lack of mKir4.1 channel block by 100 nM TPN_Q_ is also consistent with a previous study that examined the blocking effects of a TPN_Q_ derivative on Kir4.1 channels [27]. Thus, despite the positive in silico docking scores demonstrating good shape complementarity between the receptor (mKir4.1) and ligand (TPN), the affinity for TPN_Q_ block of functional Kir4.1 channels is sufficiently low, indicating the virtual screening finding represents a “false positive” that is likely due to fewer and/or weaker interaction hotspots. This structure-based result may nevertheless serve as a useful guide for re-engineering the TPN_Q_ peptide scaffold to produce stronger interactions that yield a higher-affinity TPN variant that can functionally block the Kir4.1 channel receptor given its demonstrated shape complementarity.

### 2.5. Hypothesis Testing with Structure-Based Virtual Screening

The natural biological target of TPN as an active component of bee venom is currently not known. The venom produced by stinging female worker bees is used primarily for defensive purposes from individual predation and to protect the colony from a variety of different predators and pests encountered in nature. The ability to produce an effective and deterring noxious stimulus following a sting is the result of several venom components that work in concert on diverse molecular targets expressed across a wide range of predatory species encountered in nature [12,29]. How TPN specifically might contribute towards the bee defense response is not clear, while blocking renal Kir1.1 channels and/or neuronal and cardiac Kir3.x channels in mammals seems implausible. Insects are among the many natural predators and pests that bees encounter in nature, and therefore a testable hypothesis is that certain insect Kir channels are the natural targets of TPN in bee venom.

To begin to test this hypothesis, the TPN virtual screening protocol was applied to Kir channels identified from a BLAST search of the recently reported small hive beetle genome (*Aethina tumida,* assembly Atum_1.0, NW_017852934.1) [30]. The small hive beetle is a natural parasite of *A. mellifera* and is normally maintained by guard bee aggression towards adult and beetle larva that includes envenomation [31,32]. Eight predicted *Aethina tumida* (*A. tumida*) Kir (*at*Kir) channel subunits in the genomic Reference Sequence database were identified from the BLAST search using the mouse Kir3.2 subunit sequence, with six having full-length sequence similarity to that of the TPN-sensitive mouse Kir3.2 channel (Figure 8). A multiple sequence comparison with the family of mouse Kir channel subunits indicated the *at*Kir channel subunits were most similar to each other, clustering as a distinct *at*Kir channel gene family that was closest to the mammalian Kir channel gene subfamily expressed predominantly in epithelial cells and involved in K^+^ homeostasis (i.e., Kir1.1, Kir4.x, and Kir7.1) (Figure 9B).

Structural models of the six *at*Kir channels were next constructed with homology modelling limited to the outer vestibule region as described for the mouse Kir channels. Molecular docking of TPN (conformer 13) identified two *at*Kir channels (XP_019865983.1 and XP_019865939.1) that displayed TPN docking profiles nearly equivalent to the TPN-sensitive mKir3.2 channel (3SYA.pdb) (Figure 9C,D). The interface and putative hot spots for TPN interactions with each of these *at*Kir channels were then further analyzed.

As shown in Figure 10, the predicted interface contacts between TPN and the *at*Kir channel (XP_019865983.1) outer vestibule were analogous to those between TPN-blocked mouse Kir1.1 and Kir3.2 channels, with electrostatic contacts occurring at two adjacent subunit aspartic acid residues (D113) located in the channel turret region, as well as the central “GYG” K^+^ selectivity filter at the pore entry region. Moreover, the most favored docking pose of TPN to the *at*Kir channels was comparable to the rat Kir1.1 and mouse Kir3.2 channels (cf. Figure 3), where the side chain of TPN K21 occupied the pore selectivity filter (Figure 9). Sequence comparison of the outer vestibules of TPN-sensitive mouse Kir channels with the 2 *at*Kir channels (Figure 9A) highlight the common presence of critical acidic residues in the turret region that are necessary for high-affinity TPN binding [15,26]. These in silico findings therefore support the hypothesis that TPN targets insect Kir channels, where two *at*Kir channels were identified as putative natural targets that can next be functionally tested with expression in the *Xenopus* oocyte system.

## 3. Discussion

The results of this study highlighted both the utility and the limitations of structure-based virtual screening for venom peptides and their biological targets. The similar TPN docking and interface profiles of TPN-sensitive Kir3.2 and homology-modelled Kir1.1 channels provide further evidence for the reliability of molecular docking as a first-line screening tool for TPN-sensitive Kir channels. The unexpected docking results that predicted TPN block of mKir4.1 channels indicated the virtual screening methods are sufficiently sensitive to generate “false positive” interactions, and thus have the potential to identify novel venom targets in silico for subsequent validation testing in vitro. Moreover, these finding indicated mKir4.1 channels form a favorable outer vestibule receptor for TPN interactions (i.e., shape complementarity), but lack key interface contact sites necessary for high-affinity TPN binding and channel block. This observation may aid future peptide re-engineering efforts, taking into account the unique surface chemistry of the Kir4.1 outer vestibule as either a homo-tetramer or a hetero-tetramer assembled with other Kir channel subunits (e.g., Kir4.2 or Kir5.1).

The primary discovery deploying this computational approach is the novel finding that insect Kir channels, specifically certain Kir channels expressed in *A. tumida* that are distantly related to mammalian epithelial Kir channels, may be the natural targets of the TPN venom peptide that aid in honeybee defense by lethally blocking K^+^ transport processes in insect predators. This hypothesis, assessed initially and supported using the virtual screening approach, can now focus on validation testing of the 2 specific *at*Kir channels that were identified as potentially TPN-sensitive using heterologous expression systems and functional assays. This role in bee defense would seemingly be more favorable to drive adaptive evolution of the TPN gene in *A. mellifera*, given insect predator encounters are expected to significantly outnumber encounters with mammalian predators. Other insect predators and their Kir channels of interest, including wasps and hornets, could similarly be screened using the structure-based molecular docking approach described here. Six putative *at*Kir channels were identified in this study, of those by comparison, five Kir channel subunit genes have been reported for the *Aedes aegypti* genome and three in *Drosophila melanogaster* [33,34]. The *Aedes* Kir channels are important for regulating secretion of K^+^ in Malpighian tubules and have emerged as new targets for insecticide development and mosquito control [35]. The role of *at*Kir channels in the physiology of the small hive beetle is not currently known, but may similarly regulate transepithelial secretion of K^+^ from hemolymph to urine for fluid homeostasis. 

There are inherent assumptions with the in silico structural modelling of *at*Kir channels that should be acknowledged, including less structural conservation in the transmembrane domains and large C-terminal domain and how this may affect the outer vestibule structure and TPN docking results. There also is the possible scenario that native *at*Kir channels are hetero-tetramers and would similarly impact the TPN binding landscape. The TPN conformer used for rigid-body docking was also selected based on optimal conformer docking to the mammalian Kir channels (i.e., TPN-13). It remains possible that other TPN conformations preferentially dock to insect Kir channels, where incorporating NMR ensemble docking protocols and scoring would help obviate this current bias [36]. However, despite the absence of a high-resolution TPN–Kir channel complex, the virtual screening and docking results for TPN-sensitive Kir channels consistently point to a conserved mechanism for channel block, where key electrostatic contacts with multiple channel turrets and the vestibule ring region positions the C-terminal lysine side chain of TPN to occupy the channel pore and block conduction with high-affinity binding. The novel in silico results described here establish a solid rationale for follow-up of in vitro validation assays that otherwise would not have been considered or tested. This new research direction will further address the reliability of the structural models, where future refinements to the virtual screening protocol can be introduced along with any emergent or new structural details of a high-resolution TPN–Kir channel complex. 

Screening and testing venom peptide interactions in vitro with other insect ion channel targets encountered in nature is impractical and an unlikely undertaking. The computational screening approach described here for hypothesis testing of TPN interactions with the *at*Kir channels highlights a more practical approach to identifying potential targets in silico that can then be tested in vitro. Applying this approach to a wider range of homology-modelled K^+^ channels, both homo- and hetero-tetrameric assemblies, is very feasible and could accelerate the discovery process of venom peptide targets encountered in nature.

## 4. Materials and Methods

### 4.1. Ion Channel Structures and Homology Modelling

Three high-resolution crystal structures of Kir channels were used for computationally screening venom peptide interactions: the *Gallus gallus* Kir2.2 channel (3JYC.pdb), and the *Mus musculus* Kir3.2 channel in both closed-state and pre-opened conformations (3SYA.pdb and 4KFM.pdb, respectively). For homology-modelled Kir channels, the Kir2.2 channel served as the structural template where domain modelling was restricted to the outer vestibule region (~50 amino acids) to produce a chimeric Kir channel structure using the Swiss-Model homology-modelling server as described previously [7]. The homology-modelled Kir channel subunit was then assembled as a tetramer based on the macromolecular I4 space group determined for the assembled cKir2.2 tetramer using the PDBe PISA program (Protein Interfaces, Surfaces, and Assemblies: http://pdbe.org/pisa/). All structural renderings were performed using PyMol (v1.6, Schrödinger, New York, NY, USA).

### 4.2. Computational Docking of TPN to Kir Channel Structures

All NMR solution structures of TPN (1TER.pdb) [17] were initially used for in silico docking to the Kir3.2 channel outer vestibule structures using ZDOCK 3.0.2 [25]. ZDOCK performs rigid-body searches of docking orientations between the TPN conformer and the Kir channel outer vestibule, generating 2000 complexes for each Kir channel ranked by an initial-stage scoring function that computes optimized pairwise shape complementarity, electrostatic energies, and a pairwise statistical energy potential for interface atomic contacts energies [37]. The calculated and ranked TPN docking scores were then quantitatively compared among each Kir channel tested as previously reported [7]. 

### 4.3. Kir Channel–TPN Interface Analysis

To evaluate the predicted molecular contacts between the docked TPN peptide and Kir channels, the Cluspro2.0 program was used for refined docking and RMSD greedy clustering to identify the energetically-favored complex for subsequent interface analysis [38]. The predicted interface contacts were then determined using the PDBePISA program [28,39]. PISA identifies interface contacts based on physical–chemical models of macromolecular interactions and chemical thermodynamics identified within the docked structural complexes.

### 4.4. Kir Channel Expression in Xenopus Oocytes

To test TPN_Q_ sensitivity of the Kir4.1 channel in vitro, *Xenopus* oocytes were injected with cRNA transcribed in vitro by T7 RNA polymerase (mMessage mMachine, Ambion, Austin, TX, USA) from a linearized pCMV6 vector with the mouse Kir4.1 cDNA containing a Myc-DDK tag at the C-terminus (NM_001039484, Origene Technologies, Inc., Rockville, MD, USA). For a positive assay control, some oocytes were injected with cRNA encoding the rat Kir1.1 channel of which >95% was blocked by 100 nM TPN_Q_. Oocytes were maintained for 3–5 days at 17–19 °C prior to electrophysiological recording in the following solution (in mM): 82.5 NaCl, 2.5 KCl, 1.0 CaCl_2_, 1.0 MgCl_2_, 1.0 NaHPO_4_, 5.0 HEPES, 2.5 Na pyruvate, at pH 7.5 (NaOH), with 5% heat-inactivated horse serum. Single-stage V–VI oocytes were isolated as described previously by collagenase digestion of ovarian lobes (Xenopus 1, Dexter, MI, USA) [40]. 

### 4.5. Two-Electrode Voltage Clamp Recording from Xenopus Oocytes

Macroscopic mKir4.1 and rKir1.1 channel currents were recorded by two-electrode voltage clamp recording techniques where oocytes were initially superfused with the following bath solution (in mM): 98 NaCl, 1 MgCl_2_, and 5 HEPES at pH 7.5 (NaOH). Glass electrodes having tip resistances of 0.8–1.0 MΩ were used to clamp oocytes at a holding membrane potential of −80 mV. After establishing a baseline holding current, the bath solution was changed to a “high K^+^ solution” that was comprised of an equal molar substitution of NaCl for KCl. For mKir4.1 recordings, the extracellular K^+^ concentration was 98 mM, and for rKir1.1 recordings, the concentration was 20 mM K^+^. These concentrations were determined empirically to control for peak K^+^ current amplitudes, and were attributed to differences in Kir4.1 vs Kir1.1 channel expression, single-channel conductance, and single-channel open time probability. From the −80 mV holding potential, large inward K^+^ currents were evoked and were due to the expressed Kir channels, as un-injected oocytes yielded inward currents of <100 nA (data not shown).

Rapid application and washout of 100 nM TPN_Q_ or 1 mM BaCl_2_ (dissolved in high K^+^ solutions) was performed as described previously [40]. TPN_Q_ (lyophilized solid, Tocris Bioscience, Bristol, UK) was initially dissolved in water as a 100 μM stock solution, then aliquoted and stored at −23 °C until used on the day of the experiment by diluting in the high K^+^ electrophysiological recording solution. Voltage ramps from −80 to +20 mV (200 ms in duration) were evoked periodically to assess the inward rectification characteristics of Kir channel currents during changes in the recording solutions. All recordings were performed at room temperature (21–23 °C). The membrane currents were digitized, stored, and later analyzed using an Analog-to-Digital acquisition board and PC computer (pCLAMP software, Digidata 1200 acquisition system, Axon Instruments, Foster City, CA, USA). Experiments were replicated in 7 oocytes.

## Figures and Tables

**Figure 1 toxins-11-00546-f001:**
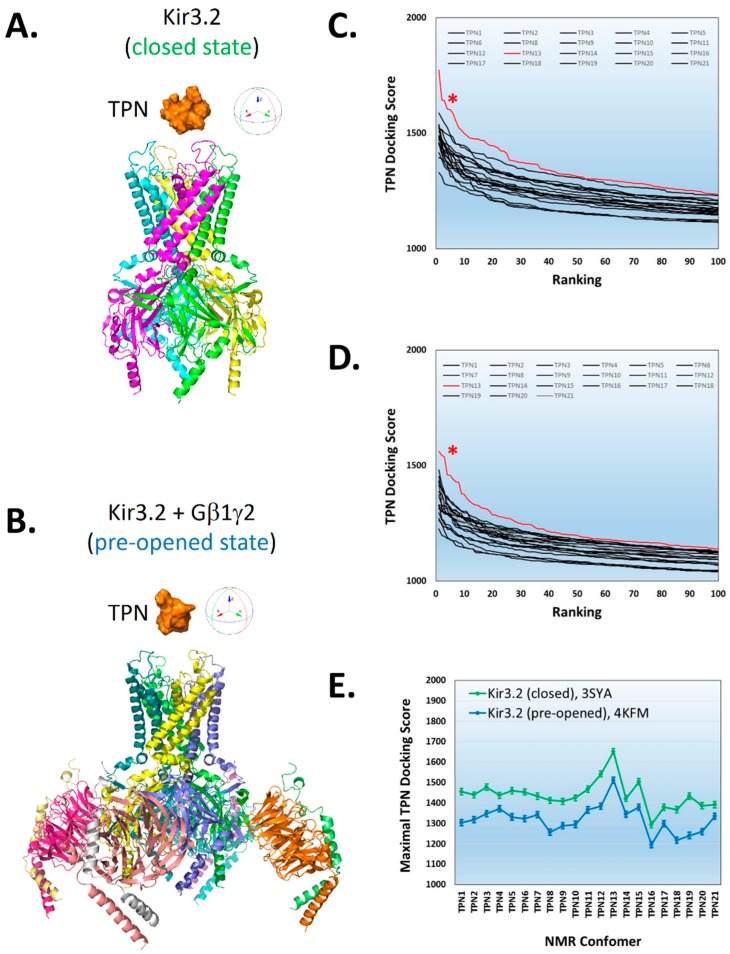
Preferential docking of the tertiapin (TPN)-13 conformer to mKir3.2 channel structures. **A.** Structural rendering of the Kir3.2 closed state (3SYA.pdb) (ribbon diagram) with TPN (orange, solid rendering) positioned above the outer vestibule. **B.** Structural rendering of the “pre-opened” conformation of Kir3.2 channel in complex with Gβγ dimers (4KFM.pdb) (ribbon diagram) with TPN (solid rendering) depicted above the outer vestibule of the channel. **C** & **D.** Plots of the top 100-ranked scores for all 21 TPN conformers docked to the Kir3.2 channel closed-state conformation (C) and pre-opened-state conformation (D). The data curves and asterisks indicated in red correspond to the TPN-13 conformer that yielded the highest score for each Kir3.2 channel conformation. **E.** Plot of the maximal docking scores (mean ± SD, top 5 complexes) for each TPN conformer docked to the Kir3.2 channel closed-state conformation (3SYA, green symbols) and pre-opened-state conformation (4KFM, blue symbols).

**Figure 2 toxins-11-00546-f002:**
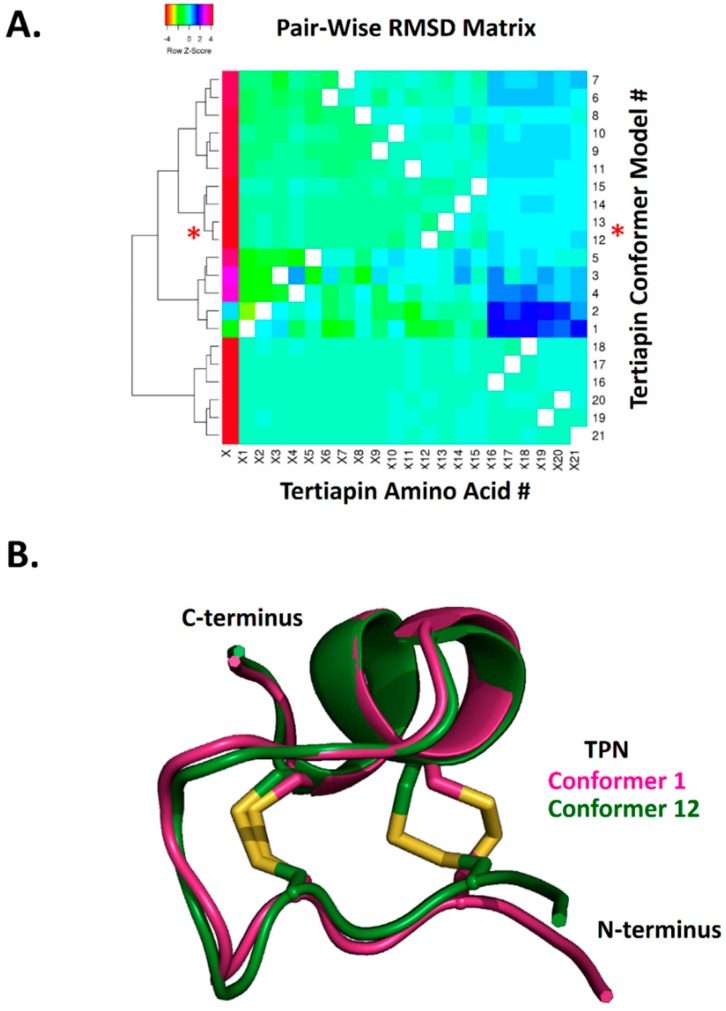
Hierarchical clustering of NMR-derived TPN structural conformers based on pairwise RMSD analysis. **A.** Heatmap and cluster analysis of the calculated pairwise RMSD along the TPN 21-amino acid alpha-carbon backbone. RMSD analysis was performed using VMD software, with the resulting 21 × 21 matrix (21 TPN conformers by 21 TPN amino acids) analyzed using Heatmapper (http://www.heatmapper.ca). The red asterisks denote the TPN-12 and TPN-13 conformer clusters that yield the top docking scores to Kir3.2 and are structurally similar. **B.** Structural alignment of TPN-1 and TPN-12 conformers, highlighting the different disulfide (Cys3–Cys14) bond configuration contributing to different peptide conformations.

**Figure 3 toxins-11-00546-f003:**
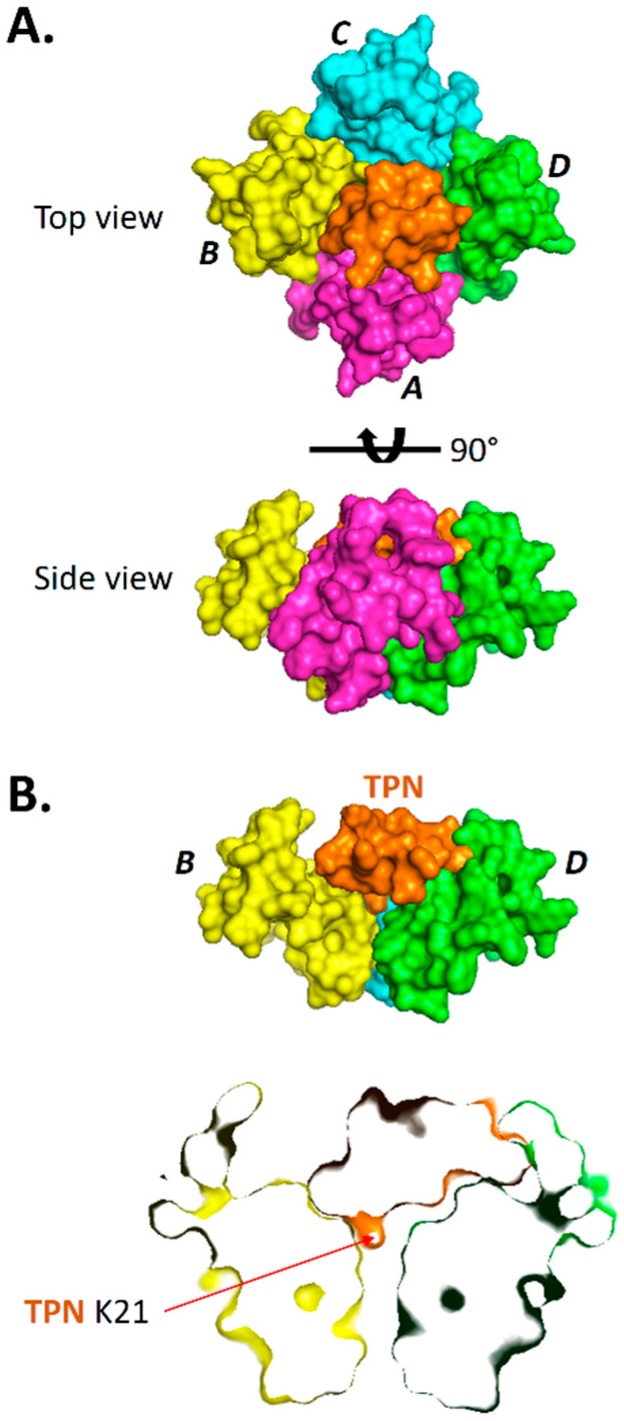
Binding pose of TPN docked to the Kir3.2 channel. **A.** Surface renderings of the TPN-13 conformer (shown in orange) docked to the Kir3.2 channel (subunits A, B, C, and D color-coded). TPN-13 was docked to the Kir3.2 channel closed state (3SYA.pdb) using Cluspro 2.0. Shown are a top view (upper image) from an extracellular vantage point, and a side view (lower image) where the TPN peptide is mostly occluded from the view by Kir3.2 subunit A (magenta). **B.** Solid side-view rendering (upper image) of the TPN-docked Kir3.2 channel with subunit A removed to expose for viewing the docked TPN-13 conformer (in orange) within the channel vestibule. The lower image is a “sectioned” side-view rendering that exposes the location of the C-terminal TPN lysine (K21) located at the mouth of the channel pore. Also visible is the juxtaposed TPN peptide with the Kir3.2 turret domain from subunit D.

**Figure 4 toxins-11-00546-f004:**
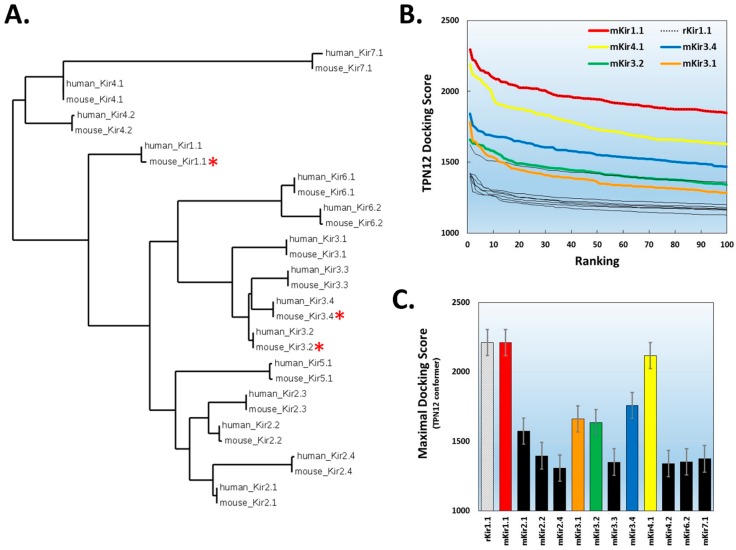
TPN docking profile to mammalian Kir channels. **A.** Dendrogram illustrating the amino acid sequence similarity of mouse and human Kir channel isoforms. A multiple sequence alignment was performed using the Constraint-based Multiple Alignment Tool (COBALT, National Center for Biotechnology Information). The tree function was used to generate the dendrogram illustrating the clustering of Kir channel subunit isoforms into their different gene subfamilies [21]. The red asterisks denote Kir channels known to be functionally blocked by TPN. **B.** Top-ranked docking scores for the TPN-12 conformer to the outer vestibule of 12 different homology-modelled mouse Kir channels. The rat Kir1.1 profile previously reported is also shown for comparison, and is identical to mKir1.1. The five Kir channels with the highest docking scores are color-coded; mKir1.1 (red), mKir4.1 (yellow), mKir3.4 (blue), Kir3.2 (green), and mKir3.1 (orange). **C.** Comparison of maximal docking scores for the TPN–Kir channel complexes. Bars represent the mean ± SD for the top five-ranked complexes for each TPN12-docked Kir channel complex. Colored bars correspond to the Kir channels color labelled in panel B.

**Figure 5 toxins-11-00546-f005:**
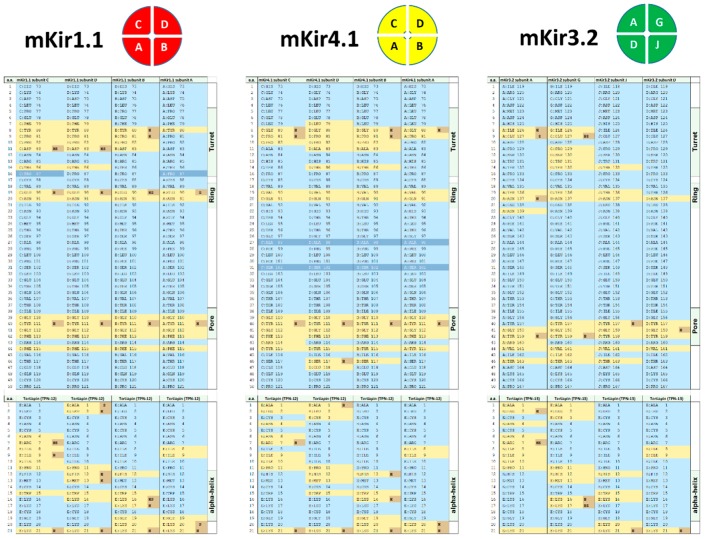
Comparative analysis of the docked tertiapin “footprint” among 3 of the highest scored Kir channel outer vestibules. Diagrams for the subunit assembled Kir channel tetramer arrangements (top view) for the homology-modelled mKir1.1 (red) and mKir4.1 (yellow) channels, and the mKir3.2 (green) homo-tetrameric channel. The predicted amino acid contacts between TPN and each Kir channel subunit are listed in each diagram below, with interfacing residues indicated in yellow and residues making hydrogen bonds or salt bridge link indicated in orange. Inaccessible residues are shown in dark blue, and solvent-accessible residues not involved in the TPN–Kir channel inferface are shown in light blue. TPN–Kir channel Interface analysis was performed using the PISA program on the top-scored docked complexes.

**Figure 6 toxins-11-00546-f006:**
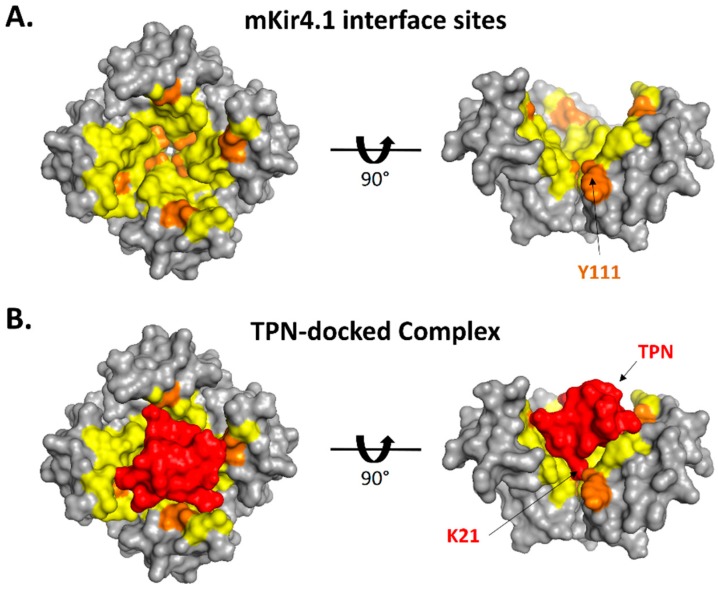
Surface interface and contact sites for the TPN-docked mKir4.1 channel complex. **A**. Surface rendering of the modelled outer vestibule of the mKir4.1 channel (top view, left; side view, right). Yellow residues depict PISA-predicted interface sites, with orange residues depicting sites with predicted H-bonds with the docked TPN peptide. For the side-view image, one of the channel subunits has been removed to expose the pore region of the vestibule. **B.** Surface rendering of TPN (red) docked to the outer vestibule of mKir4.1 channel (top view, left; side view, right). For the side-view image, the location of TPN K21 in the pore is exposed with one of the channel subunits removed.

**Figure 7 toxins-11-00546-f007:**
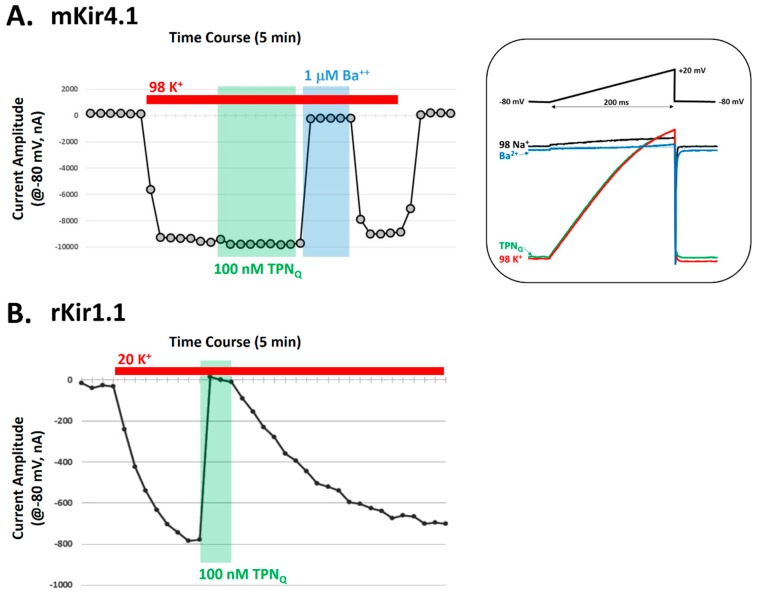
Validation testing for functional block of mKir4.1 channels by TPN_Q_. **A.** Electrophysiological recordings of mKir4.1 channel currents before and during application of TPN_Q_ in *Xenopus* oocytes. The time-course plot shows the amplitude of membrane currents at −80 mV during 2-electrode voltage clamp recording. Indicated by the red bar, high K^+^ (98 mM) application evokes inward mKir4.1-mediated K^+^ currents. Application of either 100 nM TPN_Q_ (green) or 1 μM BaCl_2_ (blue) demonstrates TPN_Q_ insensitivity and Ba^2+^ sensitivity of mKir4.1 currents. Right panel: Membrane currents evoked by the voltage-ramp protocol display the inward rectification properties of the mKir4.1 channel currents in the absence and presence of either 100 nM TPN_Q_ (green) or 1 μM BaCl_2_ (blue). The results are representative of 7 different oocytes. **B.** Positive control comparison, showing similar electrophysiological recordings of rat Kir1.1 channel currents before and during application of 100 nM TPN_Q_ in *Xenopus* oocytes. High K^+^ (20 mM) was applied to evoke inward rKir1.1-mediated K^+^ currents (red bar), where application of 100 nM TPN_Q_ (green) blocked all the Kir1.1-mediated inward currents. The results are representative of 5 different oocytes.

**Figure 8 toxins-11-00546-f008:**
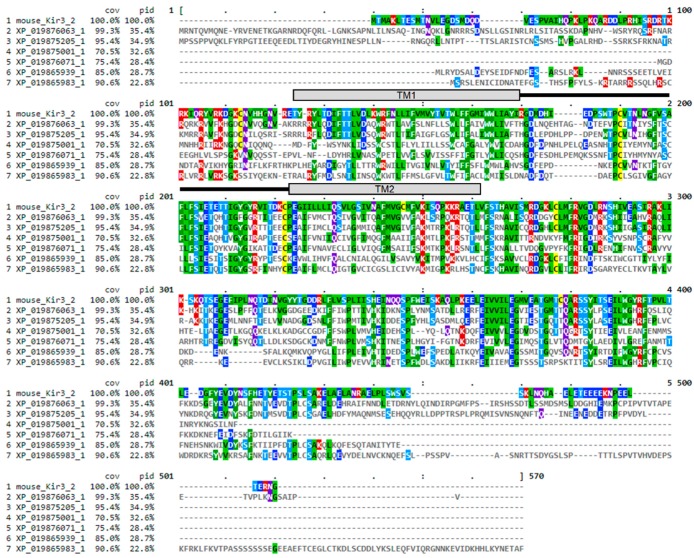
Multiple sequence alignment of six Kir channel subunit proteins identified in a BLAST search of the *Aethina tumida* (*A. tumida*) genome. The TPN-sensitive mouse Kir3.2 channel sequence was included for comparison, where the percentages of coverage (cov) and identity (pid) are shown for each *A. tumida* Kir (*at*Kir) channel subunit referenced to Kir3.2. The location of the 2 transmembrane domains (TM1 and TM2) are indicated, connected by the amino acid sequence of the the outer vestibule region that forms the receptor for TPN binding and block.

**Figure 9 toxins-11-00546-f009:**
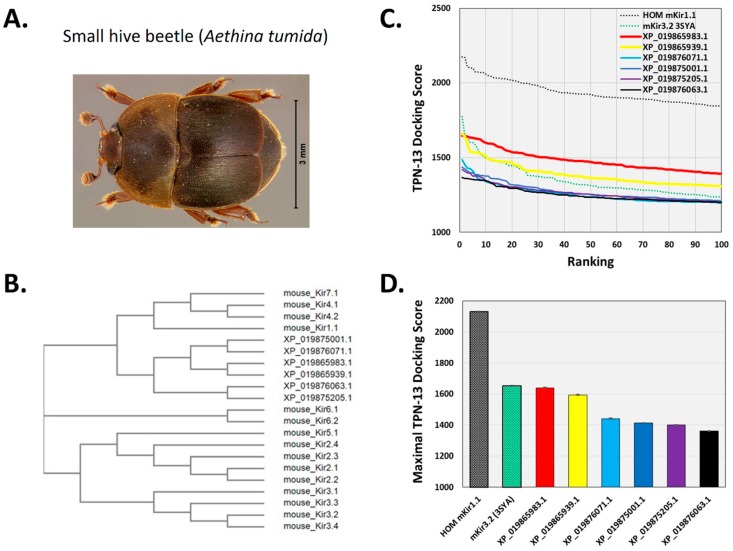
Docking TPN to homology-modelled *at*Kir channels. **A.** Photograph of the small hive beetle, *A**. tumida*, courtesy of the University of Florida, Entomology and Nematology Department. **B.** Cladogram tree from a multiple sequence alignment depicting the pairwise similarity and associated clustering among mouse Kir channel subunit protein sequences and the six identified *at*Kir channel subunits. The neighbor-joining tree was created using the Clustal Omega program at the European Bioinformatics Institute (EMBL-EBI) without distance corrections. **C.** TPN docking scores for each homology-modelled *at*Kir channel. TPN docking scores to the TPN-sensitive mouse Kir1.1 (black dotted line) and Kir3.2 channels (green dotted line) were included for benchmark comparisons. The two *at*Kir channels having TPN docking scores similar to Kir3.2 are XP_019865983.1 (red line) and XP_019865939.1 (yellow line). Molecular docking was performed using ZDOCK and the TPN-13 conformer as described in Methods. **D.** Maximal TPN docking scores from the plots in panel C are shown for comparisons in descending order. The top 5 docking scores (mean ± SD) are plotted for each homology-modelled *at*Kir channel with comparisons to those for mKir1.1 and mKir3.2.

**Figure 10 toxins-11-00546-f010:**
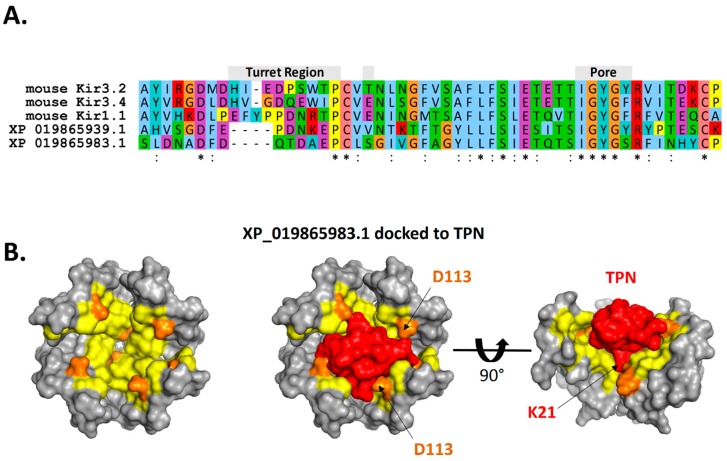
TPN docked to a homology-modelled *at*Kir channel. **A.** Sequence alignment of the outer vestibule region of TPN-sensitive mouse Kir channels and the two identified *at*Kir channels with high TPN docking scores (see Figure 8). The turret and pore regions are indicated and highlight the variable turret sequences containing acidic residues that are necessary for electrostatic contacts in TPN binding. **B.** Surface renderings of the outer vestibule of the homology-modelled *at*Kir channel XP_019865983.1. Left panel: top view where PISA-predicted TPN interface sites are indicated in yellow, and sites with predicted electrostatic contacts with the docked TPN peptide indicated in orange. Center panel: top view with the docked TPN peptide shown in red, and *at*Kir subunit turret residues (D113) indicated. Right panel: side view of the docked TPN with one *at*Kir channel subunit removed to expose the pore region where TPN K21 is predicted to make electrostatic contacts within the *at*Kir channel “GYG” K^+^ selectivity filter.

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
