# Peer review of "Identification of Aethina tumida Kir Channels as Putative Targets of the Bee Venom Peptide Tertiapin Using Structure-Based Virtual Screening Methods"

_toxins, 2019, doi:10.3390/toxins11090546_

Round 1
Reviewer 1 Report
In this study, the authors investigated the natural target for the tertiapin, a component of bee venom. The authors showed that, as has been reported previously, the bee venom component tertiapin interacts with the Kir channels. The major advance of this study is that the biological target of the tertiapin is a specific subtype of potassium channel. The analyses were mainly performed using data from the computational modeling of molecular docking, although the authors also showed the actions of tertiapin on Kir channels (Kir41 and Kir1.1) using electrophysiology techniques. The authors performed patch-clamp recordings to see the actions of tertiapin on K channels, by measuring the injecting current to hold the membrane voltage at -80mV before and after the tertiapin treatment. While the modeling and computational methodology of this study is interesting, the aim of the study is somewhat vague and the data presented in this paper are weak to lead to meaningful conclusion, or contradict the claim of the authors.
Major concerns
-The authors’ working hypothesis of interaction between tertiapin and potassium channels subtype is based on modeling analyses, such as the conformation modeling and interacting channels screening. Unfortunately, the authors' claim from the modeling does not fit well with the actual experimental data acquired using electrophysiology.
-As the experimental data contradicted the presumption based on the computational screening, more experimental data would be needed to claim the reliability and utility of the in silico methods as a useful tool for screening the biological target.
There are technical concerns regarding the electrophysiology analysis and data interpretation.
-There is no control recording data for the TPN treatment. The information for the preparation of the 100nM TPN solution should be provided in the method section, and the appropriate control experiments are needed.
-The figure 7A which used 98mM potassium solution is somewhat confusing as the authors described that they changed the bath solution to high potassium recording solution (20 mM KCl and NaCl) after the baseline recording. Why did they used the 98mM K+ solution only in Kir4.1 experiment?
-To me, it seems like the holding current is altered after the 100nM TPN treatment. In figure 7A, it goes up like 500 nA, which might be a big change.
-The authors also mentioned that they used voltage ramps protocol to assess the inward rectification of potassium current in method section, but data are missing.
Reviewer 2 Report
In this work the author(s) used structure-based computational approach and bioinformatics methods to study in silico the potential interaction of tertiapin (TPN) a small peptide present in the bee (Apis mellifera) venom with multiple inward rectifying K+ (Kir) channel of the beetle Aethina tumida a natural predator of the honey bee hive. The authors “found TPN to interact with a docking profile and interface “footprint” equivalent to known TPN-sensitive mammalian Kir channels”. Since Kir channels permeate K+ transport in cells, the authors hypothesized that tertiapin could block K+ transport via Kir channels of Aethina tumida and possibly of other insects thus serving as a mechanism to protect bee hive from predators.
Although the results shown are reliable and consistent, further in vitro experiments are still required in order to validate the observation that the reported data obtained through modelling approach by using such powerful in silico technology, could confirm the results in order to consider the interaction of tertiapin with Kir channels as a general and unfailing effect of the peptide.
Round 2
Reviewer 1 Report
Moderate editing of English language and style is required
Author Response
The language and writing of the manuscript has been reviewed and revised for improvement and overall clarity.
Thank you.
Reviewer 2 Report
The responses from the author(s) are satisfied and required additional in vitro experiments are under way by the group. Thus, the revised version is now acceptable.
Author Response
The language has been reviewed and revised for improvement and overall clarity.
Thank you.